# Beneficial Effects of Tacrolimus on Brain-Death-Associated Right Ventricular Dysfunction in Pigs

**DOI:** 10.3390/ijms241310439

**Published:** 2023-06-21

**Authors:** Asmae Belhaj, Laurence Dewachter, Astrid Monier, Gregory Vegh, Sandrine Rorive, Myriam Remmelink, Mélanie Closset, Christian Melot, Jacques Creteur, Isabelle Salmon, Benoît Rondelet

**Affiliations:** 1Department of Cardio-Vascular, Thoracic Surgery and Lung Transplantation, CHU UCL Namur, UCLouvain, 5530 Yvoir, Belgium; benoit.rondelet@chuuclnamur.uclouvain.be; 2Laboratory of Physiology and Pharmacology, Faculty of Medicine, Université Libre de Bruxelles, 1070 Brussels, Belgium; laurence.dewachter@ulb.be (L.D.); astrid.c.monier@gmail.com (A.M.); gregory.vegh@ulb.be (G.V.); christian.melot@ulb.be (C.M.); 3Department of Anatomopathology, Erasmus Academic Hospital, 1070 Brussels, Belgium; sandrine.rorive@erasme.ulb.be (S.R.); myriam.remmelink@erasme.ulb.be (M.R.); isabelle.salmon@hubruxelles.be (I.S.); 4Department of Laboratory Medicine, CHU UCL Namur, UCLouvain, 5530 Yvoir, Belgium; melanie.closset@chuuclnamur.uclouvain.be; 5Department of Critical Care, Erasmus Academic Hospital, 1070 Brussels, Belgium; jacques.creteur@ulb.be

**Keywords:** heart transplantation, right ventricular dysfunction, brain death, FK506, tacrolimus, calcineurin inhibitor, inflammation, apoptosis

## Abstract

Background: Right ventricular (RV) dysfunction remains a major problem after heart transplantation and may be associated with brain death (BD) in a donor. A calcineurin inhibitor tacrolimus was recently found to have beneficial effects on heart function. Here, we examined whether tacrolimus might prevent BD-induced RV dysfunction and the associated pathobiological changes. Methods: After randomized tacrolimus (*n* = 8; 0.05 mg·kg^−1^·day^−1^) or placebo (*n* = 9) pretreatment, pigs were assigned to a BD procedure and hemodynamically investigated 1, 3, 5, and 7 h after the Cushing reflex. After euthanasia, myocardial tissue was sampled for pathobiological evaluation. Seven pigs were used as controls. Results: Calcineurin inhibition prevented increases in pulmonary vascular resistance and RV-arterial decoupling induced by BD. BD was associated with an increased RV pro-apoptotic Bax-to-Bcl2 ratio and RV and LV apoptotic rates, which were prevented by tacrolimus. BD induced increased expression of the pro-inflammatory IL-6-to-IL-10 ratio, their related receptors, and vascular cell adhesion molecule-1 in both the RV and LV. These changes were prevented by tacrolimus. RV and LV neutrophil infiltration induced by BD was partly prevented by tacrolimus. BD was associated with decreased RV expression of the β-1 adrenergic receptor and sarcomere (myosin heavy chain [MYH]7-to-MYH6 ratio) components, while β-3 adrenergic receptor, nitric oxide-synthase 3, and glucose transporter 1 expression increased. These changes were prevented by tacrolimus. Conclusions: Brain death was associated with isolated RV dysfunction. Tacrolimus prevented RV dysfunction induced by BD through the inhibition of apoptosis and inflammation activation.

## 1. Introduction

Right ventricular (RV) dysfunction after cardiac transplantation remains a major cause of postoperative mortality [1]. In recipients, thin-walled right ventricles are often unable to adapt acutely to the high-resistance pulmonary circulation [2,3,4]. Pre-existing myocardial damage, as a result of brain death (BD) and/or ischemia–reperfusion injury, may cause graft dysfunction [5,6].

Heart transplantation faces an increasing imbalance between high-quality graft donor supply and demand. Maximizing donor heart performance is thus a crucial issue. Significant brain injury of any etiology will cause a systemic response [7], creating a pro-inflammatory environment prior to the occurrence of brain death. Brain death itself is responsible for a variety of upregulated in situ inflammatory processes and cell-damage-associated signaling pathways [6,8,9], neuro-endocrine modulations [10], hemodynamic changes [6,11,12,13], apoptosis induction [6,14], and increased release of inflammatory cytokines and stress hormones [6,11,15], which can induce direct and indirect adverse functional sequelae in right ventricle. Similarly, this has been described in heart failure [16,17].

Pulmonary hypertension and right heart dysfunction are characteristic elements of hemodynamic disturbance associated with brain death [6]. The calcineurin signaling is activated and involved in the physiopathology of pulmonary hypertension [18,19] and associated RV dysfunction [20]. Therefore, calcineurin inhibitors have been studied as a potential treatment for pulmonary arterial hypertension (PAH) in experimental models [21,22]. In addition to its immunosuppressant effects in organ transplantation, calcineurin inhibitor tacrolimus (FK506) modulates myocardial calcium homeostasis and contraction [23]. Low-dose immunosuppressant tacrolimus reversed pulmonary vascular remodeling and improved RV function in experimental models of pulmonary hypertension [24], stabilized three patients with end-stage PAH [11], and was well tolerated and safe in a larger cohort of patients with stable PAH [25].

However, whether tacrolimus has a direct effect on BD-associated RV dysfunction remains unknown because major studies were performed in recipients after a heart transplant.

There is a consensus that a better characterization and quantification of a single “primum movens” of brain-death-induced right heart injury would be of therapeutic relevance, with the goal of improving early and long-term graft function after transplantation. In this way, animal models can fill gaps in the knowledge, which are unattainable in clinical settings [26]. Our model mimicking cerebral hemorrhage is associated with right ventricular-arterial uncoupling related to the Cushing-reflex-associated catecholamine myocarditis and inflammatory and pro-apoptotic changes [6]. Neuro-humoral activation, apoptosis, as well as upregulation of pro-inflammatory cytokines are associated with increased myocardial expression of endothelial adhesion molecules and myocardial infiltration by neutrophils, which directly influence the heart failure process [6]. All these factors contribute to poor transplant outcomes, but the specific role played by immunity in early brain death has received less attention so far [27].

In this study, we explored the effects of tacrolimus as a tool to inhibit the calcineurin pathway in an experimental model of BD induced by subdural autologous blood infusion [6,28,29], with simultaneous measurements of left and right ventriculo-arterial coupling and the associated pathobiology in both ventricles.

### New and Noteworthy

This work is the first to simultaneously record the ventriculo-arterial coupling in the right and left ventricles in brain death and show isolated right heart decoupling with preserved left ventricular function.

This study is also the first to show the positive contribution of calcineurin inhibition by tacrolimus in preserving right ventricular function during brain death. Tacrolimus could therefore increase the proportion of usable hearts in BD donors by reducing right ventricular dysfunction in the recipient.

## 2. Results

At baseline, hemodynamic parameters were similar between the three groups (Figure 1 and Figure 2). One pig from the BD + Tac group died from ventricular fibrillation during blood infusion in the cerebral parenchyma.

### 2.1. Hemodynamic Evaluation (Figure 1)

BD increased HR and PAP 1 h after the CR, while SAP progressively decreased. Indexed CO increased 1 h after the CR (CR + 1 h) to decrease progressively afterward and reach the basal value at CR + 7 h. Pulmonary vascular resistance (PVR) progressively increased with an approximately tripled value at CR + 7 h compared to baseline. LAP and RAP also increased, with occluded PAP remaining below 12 mmHg throughout the protocol. Systemic vascular resistance (SVR) decreased from CR + 1 h, with increasing need for NA up to 0.13 µg.kg^−1^.min^−1^ at CR + 7 h. 

Tacrolimus completely prevented changes observed in HR, CO, PVR, SAP, SVR, and RAP at CR + 7 h and limited the increase in PAP without affecting LAP. The need for NA administration was also significantly less important after tacrolimus.

### 2.2. Tacrolimus Pretreatment Prevents Right Ventriculo-Arterial Uncoupling (Figure 2)

BD was associated with a progressive decrease in RV Ees/Ea, suggesting progressive RV-pulmonary arterial uncoupling after the CR. At CR + 1 h, RV systolic function remained adapted to its afterload, as indicated by proportional increases in Ees and Ea, but at CR + 7 h, BD resulted in an insufficient increase in RV systolic function before a marked increase in RV afterload, which resulted in a profound decrease in Ees/Ea, suggesting RV-pulmonary arterial uncoupling. In contrast, the LV Ees-to-Ea ratio remained unchanged throughout the experiment.

Tacrolimus completely prevented RV-pulmonary arterial uncoupling observed after BD. Indeed, RV afterload remained unchanged after tacrolimus pretreatment, with adapted RV systolic function throughout the protocol. LV-arterial coupling was not altered by tacrolimus.

### 2.3. Circulating Cardiac Markers (Figure 3)

Serum levels of C-reactive protein, troponin-I, and creatinine remained stable throughout the protocol. BD increased serum urea levels. Nevertheless, all these plasma levels were already within the normal range for pigs in the three groups. After tacrolimus, there was no change in these circulating marker levels compared to the BD group.

### 2.4. Tacrolimus Prevents RV and LV Activation of Apoptosis (Figure 4)

BD was associated with Bax-to-Bcl-2 and Bax-to-Bcl-XL gene ratios in the RV and LV. To confirm the activation of apoptotic processes, TUNEL staining was performed to evaluate the myocardial apoptotic rate. BD was associated with three-fold increases in RV and LV apoptotic rates at CR + 7 h.

Tacrolimus prevented BD-induced Bax−to−Bcl−2 and Bax−to−Bcl−XL ratios overexpression, as well as the increase in the apoptotic rate in the RV (totally) and LV (partly).

The RV Ees/Ea ratio was inversely correlated with the relative gene expression of Bax−to−Bcl−2, Bax−to−Bcl−XL ratios, and the apoptotic rate in the RV.

### 2.5. Tacrolimus Pretreatment Alters the Expression of Inflammatory Regulators and Cytokines (Figure 5 and Figure 6)

As illustrated in Figure 5, BD was associated with increases in RV pro-inflammatory IL-6-to-IL-10, associated receptors IL-6R-to-IL-10R ratios, and IL-6 signal transducer Gp130 gene expression. BD was associated with increased RV gene expression of IL-1β and its antagonist receptor IL-1RN. No change in iNOS mRNA level was observed in the RV after BD, while HO-1 expression increased. BD was associated with increased expression of VCAM-1 in the RV and LV (*p* = 0.06), while the expression of ICAM-1 and ICAM-2 remained unchanged. In the LV, the IL-6R mRNA content, the IL-6R-to-IL-10R ratio and Gp130 (*p* = 0.07), HO-1, and VCAM-1 (*p* = 0.06) increased after BD, with no changes in IL-1β and its antagonist receptor IL-1RN, IL-6, and IL-10, Gp130, TNF-α, iNOS, ICAM-1, and ICAM-2.

BD-induced RV dysfunction was associated with an increased number of extravascular neutrophils per mm^2^ in the RV and LV (Figure 6).

In pigs with BD-induced RV dysfunction, tacrolimus prevented the increase in RV pro-inflammatory IL-6-to-IL-10 and IL-6R-to-IL-10R ratios and IL-6 signal transducer Gp130, IL-1 receptor antagonist IL-1RN, iNOS, and VCAM-1 gene expression, while the expression of IL-1β and HO-1 remained high in the RV (Figure 5). After tacrolimus, changes in gene expression observed in the LV were similar to those observed in the RV (Figure 5). After pretreatment, the number of extravascular neutrophils per cm^2^ decreased in the RV and LV, but they remained higher than the myocardial neutrophil infiltrate observed in controls (Figure 6).

The RV Ees/Ea ratio was inversely correlated with the gene expression of IL-1β (Figure 5) and its receptor IL-1RN (correlation not shown), pro-inflammatory IL-6-to-IL-10 (Figure 5), and associated receptors IL-6R-to-IL-10R ratios (correlations not shown), VCAM-1 (Figure 5), and iNOS (Figure 5), as well as with the number of extravascular neutrophils per cm^2^ in the RV (Figure 6).

### 2.6. Altered Myocardial Expression of β-Adrenergic Receptors Was Prevented by Tacrolimus (Figure 7)

BD-induced RV dysfunction was associated with decreased β1- and increased β3-adrenoreceptor gene expression, while β2-receptor expression remained unchanged in RV. In the LV, β1-adrenoreceptor mRNA content decreased, while β2-adrenoreceptor mRNA level increased with no change in β3-receptor expression.

Tacrolimus prevented all changes observed in β-adrenoreceptor mRNA expression in the RV and LV.

The RV Ees/Ea ratio was positively correlated with gene expression of b1-adrenoreceptor and inversely related to gene expression of β3-receptor expression in the RV.

### 2.7. Tacrolimus Partly Prevented the Altered Expression of Myocardial Calcium-Handling Molecules (Appendix A)

To decipher the molecular mechanisms underlying BD-induced RV dysfunction, we characterized calcium-handling molecules implicated in the cardiac contraction/relaxation cycle. BD was associated with decreased RV expression of SERCA2A (*p* = 0.003), PLB (*p* = 0.084), and RyR2, all implicated in calcium translocation from the cytosol into the sarcoplasmic reticulum, with decreased expression of CACNA1C, a voltage-dependent calcium channel mediating calcium influx into the cell. BD-induced RV dysfunction was also associated with decreased expression of calcium/calmodulin-dependent protein kinase CAMKIId (*p* = 0.0001), while the RV expression of the SERCA2-to-PLB ratio and Na^+^-calcium exchanger NCX1 implicated in cardiac relaxation did not change. In the LV, the gene expression of RyR2, CACNA1C, and CAMKIId (*p* = 0.0001) decreased, with no changes in SERCA2A, PLB, and NCX1 gene expression after BD (Appendix A).

The RV Ees/Ea ratio was positively correlated with the gene expression of CAMKIId (R = 0.5036, *p* < 0.01) in the RV.

### 2.8. Myocardial Expression of Cardiac Hypertrophic Markers (Figure 7)

We assessed the gene expression of recognized hypertrophic markers and found that the RV gene expression of NPPA (encoding ANP), and MYH7 (encoding β-myosin heavy chain (MHC); *p* = 0.00001) decreased in pigs with BD-induced RV dysfunction, while the RV expression of NPPB (encoding BNP) increased. No changes were noticed in the RV MYH6 (encoding a-MHC) expression and in the MYH7-to-MYH6 ratio. In the LV, BD induced decreased gene expression of MYH6 (*p* = 0.01) and MYH7 (*p* = 0.0004) and increased NPPB gene expression, while ACTA1 and NPPA did not change.

Tacrolimus did not impact altered RV ad LV gene expression of cardiac hypertrophic markers compared to those observed in non-treated animals, except for RV MYH6 gene expression, which further decreased after tacrolimus pretreatment (*p* = 0.04). The MYH7-to-MYH6 ratio was also increased in the RV after tacrolimus. 

The RV Ees/Ea ratio was correlated with the gene expression of the MYH7-to-MYH6 ratio (Figure 7) in the RV.

### 2.9. Tacrolimus Partly Prevented the Myocardial Altered Expression of eNOS and Central Regulators of Cardiac Metabolism (Figure 7)

Multiple studies signify the cardioprotective role of eNOS in various cardiovascular diseases [30]. BD-induced RV dysfunction was associated with increased gene expression of eNOS in the RV and LV. Pretreatment with tacrolimus prevented this increased eNOS expression. The RV Ees/Ea ratio was inversely correlated with the gene expression of eNOS in the RV.

In the adult heart, fatty acids are known as the predominant fuel, and the heart switches its substrate preference toward glucose during stress conditions. After BD, the RV and LV expression of GLUT1 increased, while GLUT4 expression decreased in the RV with no change in GLUT4 expression in the LV. Expression of CD36, a surface receptor regulating myocardial fatty acid uptake, was decreased in the RV and LV after BD.

Tacrolimus partly prevented the increase in GLUT1 and the decrease in GLUT4 expression induced by BD but failed to restore RV and LV CD36 expression.

The RV Ees/Ea ratio was inversely correlated with the gene expression of GLUT1 in the RV and LV and positively related to the gene expression of GLUT4 in the RV.

### 2.10. Myocardial Expression of Toll-Like Receptor (TLR) Signaling Pathways (Figure 8)

Because TLRs have been incriminated in inflammation-associated cardiac dysfunction [31], we explored TLR expression. BD was associated with decreased mRNA content of TLR2 and the inflammasome component NLRP3, with increased TLR4 expression and no change in TLR9 expression in the RV. In the LV, NLRP3 expression decreased, while TLR4 expression increased with no change in TLR2 and TLR9.

Tacrolimus failed to prevent the changes observed in TLR2 and TLR4 expression, prevented decreased NLRP3 expression, and increased TLR9 expression.

## 3. Discussion

In the present study, we found that tacrolimus prevented RV-arterial uncoupling, the associated RV activation of apoptotic and inflammatory processes, and altered the expression of molecules implicated in calcium-dependent myocardial contraction in an experimental porcine model of BD. LV-arterial coupling was not altered after BD and remained preserved after tacrolimus pretreatment.

An acute increase in intracranial pressure and subsequent BD have been shown to cause a marked increase in pulmonary artery pressure and pulmonary vascular resistance usually associated with a higher RV contractile state causing an increase in RV pump function [6]. Similar changes were observed in the LV, with an acute increase in cardiac output and artery pressure usually leading to tachycardia and LV hypercontractility [11,32,33,34,35,36]. These responses reflect sympathetic nervous discharge [37,38,39], with high plasma catecholamine levels [40]. All these changes were observed in our experimental model of BD in pigs [6,28]. Simultaneous measurements of RV and LV functions were performed. Analyzing the Ees-to-Ea ratio showed that, in contrast with preserved LV-artery coupling, the RV was not able to adapt its systolic function to its increased afterload during the acute phase after BD, suggesting that the RV was more vulnerable. Abnormal sympathetic tone activation, reflected by an elevated heart rate, could contribute to its reduced reserve to further adapt. Pretreatment with tacrolimus was able to preserve RV-pulmonary arterial coupling after BD, mainly acting on RV afterload, as shown previously with preventive corticoid therapy [6]. Indeed, pulmonary vascular resistance and RV pulmonary Ea were reduced, while RV contraction (evaluated by RV Ees) did not proportionally increase to maintain the Ees-to-Ea ratio.

RV dysfunction in brain-dead donors was related to a RV myocarditis caused by massive catecholamine release [6]. However, the pathobiological crosstalk between the RV and LV pumps has been insufficiently described [41]. Previous studies showed a significant increase in systemic vascular resistance during the acute phase after BD [11,33,42], while pulmonary vascular resistance was not altered [33,40] or increased [11,35,36,42,43]. In the present study, systemic arterial pressure decreased while pulmonary artery pressure and pulmonary vascular resistance rapidly increased after BD. The left atrial pressure estimated by the pulmonary arterial wedge pressure rose slowly during the experiment but remained in a normal range. Altogether, this suggested that RV dysfunction was not related to LV diastolic dysfunction but rather to a direct effect on the RV. The present experimental model of BD in pigs provides clinically useful information concerning potential clinically relevant therapeutic interventions with potential directly targeted RV function preservation in BD.

We previously experimentally studied the hemodynamic response to a comparable acute increase in RV afterload and showed that RV dysfunction was mainly related to RV activation of apoptosis and inflammatory processes [6,43,44]. Various explanations could be given for the insufficient ability of the RV to adapt via inotropic mechanisms. BD-induced RV injury and subsequent cardiomyocyte apoptosis may play a role in the insufficient RV myocardial contractile response. Histological analyses showed increased cardiomyocyte apoptosis in the RV and LV of brain-dead hearts associated with decreased gene expression of major determinants of calcium-dependent myocardial contraction, including SERCA2A, Ryr2, CACNA1C, and CAMKIIδ. Interestingly, apoptosis and altered expression of calcium^+^-dependent myocardial contraction determinants were highly correlated with RV uncoupling. This could, at least partly, explain the impaired ability of brain-dead RV to compensate for the acute increase in RV afterload via positive inotropy. We can presume that this was due to the characteristic thin-walled structure of the RV being less able to adapt to stress and secondarily to the rapid and massive increase in RV afterload, while the LV afterload decreased. In clinical heart transplantation, tacrolimus prevented apoptosis and decreased the expression of myocardial calcium-handling molecules, which may have some relevance for graft preservation in clinical heart transplantation. Mechanistically, tacrolimus binds the 12-kDa FK506-binding protein (FKBP12) and thereby increases BMP downstream in pulmonary arterial endothelial cells [24]. Tacrolimus require activin receptor-like kinase 1 (ALK1) as a BMP-Receptor-2 co-receptor to reduce collagen production in cardiac fibroblasts. Interestingly, FKBP12 itself is associated with the calcium release channel of cardiac muscle, and pharmacological dissociation of this complex alters its gating characteristics [45,46], causing increased calcium accumulation in the cardiomyocyte and a potential additional positive inotropic effect on tacrolimus [47,48].

After BD, central nervous system injury is associated with a massive inflammatory response syndrome, occurring early after injury; it is responsible for the activation of inflammatory processes in organs. In the present study, the inflammatory markers were markedly upregulated in both the RV and LV after BD, with increased expression of inflammatory cytokines IL-1β, the IL-6-to-IL-10 ratio, and associated receptors and adhesion molecule VCAM-1, together with myocardial neutrophil infiltration. This is consistent with the global inflammatory response syndrome observed after BD and reinforces the relevance of testing the preventive strategies targeting inflammatory and immune pathways to preserve RV function in BD. In addition, TLRs, which are able to generate innate immune responses, have been implicated in cardiac dysfunction [31]. TLR4, whose levels are the highest compared with other TLRs in the heart, plays a critical role in myocardial inflammation and the associated cardiac dysfunction. In this study, the cardiac expression of TLR4 increased, which probably promoted inflammation and exacerbated cardiac dysfunction. Abnormal inflammatory and immune status can cause cardiac subcellular component abnormalities, including oxidative stress, mitochondrial dysfunction and associated cell death, and impaired calcium handling, leading to impaired myocardial contraction [49]. Tacrolimus reduced the RV (cytokine expression, VCAM-1, and iNOS were correlated with RV Ees/Ea) and LV activation of inflammatory processes but failed to reduce TLR4 expression.

Myocardial hypertrophy and fibrosis often observed in transplanted human hearts are known to have adverse effects on long-term cardiac function [50]. Cardiac hypertrophy and remodeling are also crucial early adaptative steps (before becoming pathological), which are accompanied by the re-expression of the fetal gene program comprising NPPA, ACTA1, and MYH7 [51,52]. In this study, the RV expression of these genes was decreased, which likely contributed to the early insufficient adaptation of mechanical and contractile properties of the RV to increased mechanical stress. Additionally, maladaptive energetic use plays an important role in the pathophysiology of the failing heart contributing to reduced functional capacity [53]. Here, we found BD-induced cardiac metabolic derangements with decreased expression of fatty acid transporter CD36 and major cardiac glucose transporter GLUT4, with increased expression of glucose transporter GLUT1 in both the RV and LV. Tacrolimus partly prevented these metabolic changes, which likely contributed to preserved cardiac function efficiency.

In conclusion, tacrolimus prevented pulmonary hypertension and RV dysfunction observed after brain death, with the associated reduced activation of apoptosis, inflammatory, and immune systems. Tacrolimus should therefore be considered as a potential therapeutic agent, which can improve cardiac graft function and minimize the deleterious impact of brain death after cardiac transplantation.

## 4. Materials and Methods

This study was conducted in accordance with the Guiding Principles in Care and Use of Animals of the American Physiologic Society and was approved by the Institutional Ethics Committee on Animal Welfare of the Faculty of Medicine of the Université Libre de Bruxelles (Brussels, Belgium; protocol number: extended 510N). Animals were delivered by an approved breeder (Jan Derook Farm, Belgium).

The day before the experiment, twenty-four pigs (mean weight, 51 ± 1 kg; range, 43–59 kg) were randomly assigned to a control group (*n* = 7) or to receive a BD procedure (*n* = 17). BD-randomized animals were randomly assigned to placebo (BD group; *n* = 9) or 0.25 mg.kg^−1^ tacrolimus (PROGRAF^®^, Astellas, Tokyo, Japan; BD + Tac group; *n* = 8) administered orally with food twice a day.

### 4.1. Animal Preparation

As previously reported [6,28,29,54,55], the animals were premedicated with intramuscular 20 mg·kg^−1^ ketamine and 0.1 mg·kg^−1^ midazolam, anesthetized with intravenous 1 mg.kg^−1^ midazolam and 15 mg.kg^−1^ remifentanyl, and paralyzed for thoracotomy with intravenous 0.2 mg·kg^−1^·h^−1^ rocuronium bromide. Anesthesia was maintained with intravenous continuous infusion of 0.1 mg.kg^−1^·h^−1^ midazolam and 7.5 mg·kg^−1^·h^−1^ remifentanyl. Animals were ventilated with an inspired oxygen fraction (FiO_2_) of 0.4–1.0 to maintain an arterial oxygen saturation (SaO_2_) > 90%, a respiratory rate of 12–20 breaths·min^−1^, and a tidal volume of 15–25 mL·kg^−1^ to achieve an arterial PaCO_2_ between 35 and 45 mmHg, and positive end-expiratory pressure between 5 and 8 cm H_2_O.

A flow-directed balloon-tipped pulmonary artery catheter (131H-7F; Baxter-Edwards, Irvine, CA, USA) was inserted via the left external jugular vein and positioned in the pulmonary artery to measure pulmonary artery pressure (PAP), right atrial pressure (RAP), thermodilution cardiac output (CO), and central body temperature, and for mixed venous blood sampling. A polyethylene catheter was inserted in the thoracic aorta via the right carotid artery to measure systemic artery pressure (SAP) and left atrial pressure (LAP) and to sample arterial blood. A left thoracotomy was performed, and an ultrasonic flow probe (T101; Transonic Systems, Ithaca, NY, USA) was placed around the main pulmonary artery and aorta to measure the instantaneous pulmonary blood flow (Q); high-fidelity 5F manometer-tipped catheters (Millar Instruments, Houston, TX, USA) were introduced in the right and left ventricles and in the pulmonary artery and aorta to measure instantaneous ventricular flow outputs and compute instantaneous PAP and SAP.

Balanced crystalloid and modified liquid gelatin solutions were perfused at 10 mL.kg^−1^.h^−1^ to maintain an RAP of 6–8 mmHg. If SAP was below 65 mmHg with a maximal RAP value of 10 mmHg, a continuous infusion of norepinephrine was started to maintain blood pressure and titration [56]. Heart rate (HR), PAP, and SAP were continuously monitored. Arterial blood gases were controlled every 30 min.

### 4.2. Brain Death Procedure

After providing anesthesia to the animals and placing the monitoring equipment, one hole was drilled in the temporoparietal cranium. Autologous blood was slowly infused (0.5 mL·min^−1^; 240 min) in the parenchyma through an 18-gauge SURFLO^Ò^ catheter (TERUMO^®^, SR-OX1851CA) subdurally inserted to induce BD and to record instantaneous intracranial pressure. Approximately 1 h after the Cushing reflex (CR) response, which lasted 30–60 min, BD was objectivated with standard criteria, including: (a) deep coma, excluding reversible factors; (b) absence of oculo-pupillary and corneal reflexes; (c) positive apnea testing; (d) no increase in HR after administration of atropine (1 mg) when monitored for 5–15 min; and (e) disappearance of the “tic reaction” [6,28,29,57,58]. When the CR was objectivated, anesthesia and paralysis drugs were stopped. Then, protective ventilation was applied with a tidal volume of 6 mL·kg^−1^ and targeted plateau pressures below 30 cm H_2_O. Periodic deep inspirations were administered to prevent atelectasis.

### 4.3. Data Acquisition and Analysis

Each data set included blood gas analysis; measurements of SAP, PAP, LAP, RAP, and CO, using a Biobox^®^ acquisition system (Biomedisoft^®^, Brussels, Belgium); systemic arterial blood sampling; and lung biopsies, as previously reported [6,28,29]. After 20 min of post-BD stabilization, HR, SAP, and PAP data were collected at baseline at 1 h (CR + 1 h), 3 (CR + 3 h), 5 (CR + 5 h), and 7 h (CR + 7 h) after the CR. Afterward, the animals were euthanized by pentobarbital sodium (Nembutal^®^, Oak, Lake Forest, IL, USA) overdose, and cardiac tissue (RV and LV free walls) was sampled. Transmural myocardial sections were immediately snap-frozen and stored at −80 °C for pathobiological evaluation, and, after overnight fixation, they were embedded in paraffin for immunohistochemistry. Control animals were euthanized after baseline data set acquisition.

Instantaneous pulmonary and aortic pressures and flow waves were sampled at 200 Hz and analyzed with data-analysis software 1.0 (Biomedisoft^®^, Brussels, Belgium). Five consecutive end-expiratory heartbeats were analyzed for each data set. The “single-beat method” was used to evaluate RV- and LV-arterial couplings, computing the end-systolic elastance (Ees) as the slope of end-systolic pressure–volume relationship, the effective artery elastance (Ea) as the slope of end-diastolic–to-end-systolic relationship, and the coupling efficiency as the Ees/Ea ratio [6,59].

### 4.4. Biological and Histological Assessment

Real-time quantitative polymerase chain reaction (RT-qPCR)

Relative mRNA expression was evaluated using RT-qPCR experiments according to a previously reported protocol [6,28,54,60] for *sus scrofa* pro-apoptotic Bax and anti-apoptotic Bcl-2 and Bcl-XL mitochondrial membrane transport regulators, interleukin (IL)-1β and its receptor antagonist (IL-1RN), IL-6, its receptor (IL-6 R) and its signal transducer GP130, IL-10 and its receptor (IL-10 R), tumor necrosis factor-α (TNF-α), heme oxygenase (HO)-1, inducible nitric oxide synthase (iNOS, also called NOS2), intercellular adhesion molecules (ICAM)-1 and -2, vascular cell adhesion molecule (VCAM)-1, β-1, -2 and -3 adrenergic receptors, sarcoplasmic/endoplasmic reticulum Ca^2+^-ATPase 2a (SERCA2A), phospholamban (PLB), ryanodine receptor 2 (RyR2), voltage-gated Ca^2+^ channel subunit alpha1 C (CACNA1C), Na^+^/Ca^2+^ exchanger 1 (NCX1), Ca^2+^/calmodulin-dependent protein kinase II, delta (CaMKIIδ), α-1 skeletal muscle actin (ACTA1), α- and β-myosin heavy chains (MYH6 and MYH7), natriuretic peptide A (NPPA) and brain natriuretic peptide (NPPB), endothelial NOS (eNOS, also called NOS3), glucose transporters (GLUT) 1 and 4, thrombospondin receptor CD36 (CD36), Toll-like receptors (TLR-) 2 and 4, and NLR family pyrin domain containing three (NLRP3) mRNA sequences (Table 1). Amplification of residual genomic DNA was avoided by selecting intron-spanning primers when exon sequences were known, and an analysis was performed to verify that the primer pairs only matched the sequence of interest. For each sample, the amplification reaction was performed in triplicate using SYBR-Green PCR Master Mix (Quanta Biosciences, Gaithersburg, MD, USA), specific primers, and diluted template cDNA. The result analysis was performed using the iCycler System (Bio-Rad Laboratories). Relative mRNA quantification was achieved using the 2^−∆∆Ct^ method [61] by normalization, with the ribosomal protein L4 (RPL4; Table 1) used as a housekeeping gene.

### 4.5. Immunohistochemistry: Terminal Deoxynucleotidyl Transferase dUTP Nick-End Labeling (TUNEL) Staining to Assess Myocardial Apoptosis

Cardiomyocytes undergoing apoptosis were detected by TUNEL staining using the ApopTag^®^ Plus Peroxidase In Situ Apoptosis Detection Kit (EMD Millipore, Temecula, CA, USA), as previously described [6,62]. Negative controls run without the TdT enzyme and positive controls pretreated with DNase were tested. For each tissue specimen, twenty randomly chosen fields were examined. The myocyte apoptotic rate was calculated as the ratio of apoptotic cells (TUNEL-positive or brown nuclei) to total cells (brown + blue nuclei) (×100 to be expressed as a percentage). Two independent investigators performed all counts in a blinded manner. The mean value was used for analysis. 

### 4.6. Immunohistochemistry: Evaluation of Myeloperoxidase-Positive Inflammatory Cell Infiltrate 

Myocardial-infiltrating neutrophils were myeloperoxidase (MPO)-stained, as previously described [6,44]. Myocardial sections of 3 μm were incubated in an antigen unmasking solution (Vector Laboratories, Burlingame, CA, USA) and heated in a bath for 20 min at 95 °C for antigen retrieval. Endogenous peroxidase activity was quenched with hydrogen peroxide in phosphate-buffered saline (PBS; 5%) for 10 min, and the sections were blocked by incubation with bovine serum albumin (BSA) in PBS (1%) for 60 min. Sections were allowed to react overnight at 4 °C with polyclonal rabbit anti-human myeloperoxidase (MPO; AB-9535, Abcam, Cambridge, UK; 1:100 diluted in PBS with 1% BSA). Sections were then incubated with biotinylated anti-rabbit IgG (BA-1000, Vector Laboratories, Burlingame, USA; 1:200 diluted in PBS with 1% BSA) and subsequently with streptavidin–peroxidase (SK-4100; Vector Laboratories). Antibody binding was detected with a liquid diaminobenzidine substrate kit (Vector Laboratories). The appearance of a brown reaction product was observed under a light microscope. Nuclei were counterstained with hematoxylin (Sigma-Aldrich, St. Louis, MO, USA) and mounted. Negative controls run without the primary antibody were tested and found to be negative. In each tissue slide of the RV and the LV, 15 microscopic intramyocardial fields (light microscopy, original magnification ×400) were randomly chosen. In these fields, the numbers of extravascular neutrophils (MPO-positive) were counted. Intravascular inflammatory cells were excluded. The total surface of each sample was measured using ImageJ software (National Institutes of Health). The average numbers of intramyocardial extravascular inflammatory cells per µm^2^ were then calculated as the total score for each specimen. Cells were counted by two investigators who were blinded to the assigned group.

### 4.7. Plasma Levels of Inflammatory, Cardiac, and Kidney Biomarkers

All serum measurements were performed using the Vitros^®^ 5600 Integrated System (Ortho Clinical Diagnostics, Singapore). Serum levels of C-reactive protein (CRP) were evaluated using the immunoturbidimetric method (DIAGAM^®^ C-reactive protein Gold Latex). The serum concentration of highly sensitive cardiac troponin I was measured by an immunometric immunoassay (VITROS^®^ Immunodiagnostic Products HS Troponin I) using a chemiluminescent reaction. Serum urea and creatinine levels were measured using the enzymatic method (VITROS^®^ Chemistry Products UREA Slides) with spectrophotometric detection. 

### 4.8. Statistical Analysis

Analyses were performed using PRISM^®^ version 5 (GraphPad Software Inc., San Diego, CA, USA; www.graphpad.com). All data are expressed as mean ± standard error of the mean. Data were analyzed using a two-factor (group × time) repeated measures analysis of variance. When the F ratio reached a critical value <0.05, Fisher’s exact *t*-tests were performed with a Bonferroni correction for multiple comparisons. *p*-values < 0.05 were considered statistically significant. 

Using univariable and multivariable analyses with Cox proportional hazard model linear regression analyses, we tested the relationships of the biological variables for apoptosis, inflammation, ventricular function, and myeloperoxidase-positive inflammatory cell infiltrate with right and left ventriculo-arterial couplings. 

Asmae Belhaj had full access to all the data in the study and takes responsibility for its integrity and the data analysis.

## Figures and Tables

**Figure 1 ijms-24-10439-f001:**
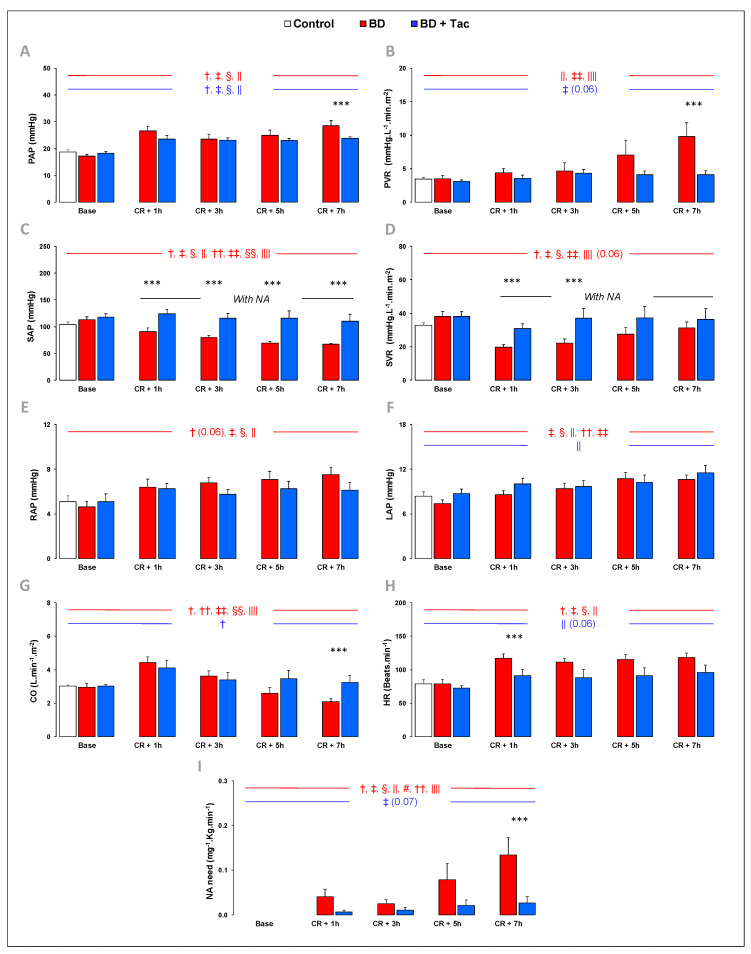
Pulmonary and systemic hemodynamics were evaluated at baseline, 1 (CR + 1 h), 3 (CR + 3 h), 5 (CR + 5 h), and 7 h (CR + 7 h) after Cushing reflex (CR) in the control (*n* = 7; white bars), placebo (brain death, BD; *n* = 9; red bars), and tacrolimus pretreated brain death (BD + Tac; *n* = 8; blue bars) groups. PAP (**A**) indicates mean pulmonary arterial pressure; PVR (**B**), pulmonary vascular resistance; SVR (**C**), systemic vascular resistance; SAP (**D**), mean systemic arterial pressure; RAP (**E**), right atrial pressure; LAP (**F**), left atrial pressure evaluated by the pulmonary arterial wedge pressure; CO (**G**), cardiac output; HR (**H**), heart rate; and NA need (**I**), noradrenaline need. Values are expressed as mean ± standard error of the mean. *** *p* < 0.05 BD vs. BD + Tac, † *p* < 0.05 Base vs. CR + 1 h, ‡ *p* < 0.05 Base vs. CR + 3 h, § *p* < 0.05 Base vs. CR + 5 h, || *p* < 0.05 Base vs. CR + 7 h, # *p* < 0.05 CR + 1 h vs. CR + 3 h, †† *p* < 0.05 CR + 1 h vs. CR + 5 h, ‡‡ *p* < 0.05 CR + 1 h vs. CR + 7 h, §§ *p* < 0.05 CR + 3 h vs. CR + 5 h, |||| *p* < 0.05 CR + 3 h vs. CR + 7 h. The comparisons between the different recording or sampling times are marked in red for the “BD” group and in blue for the “BD + Tac” group.

**Figure 2 ijms-24-10439-f002:**
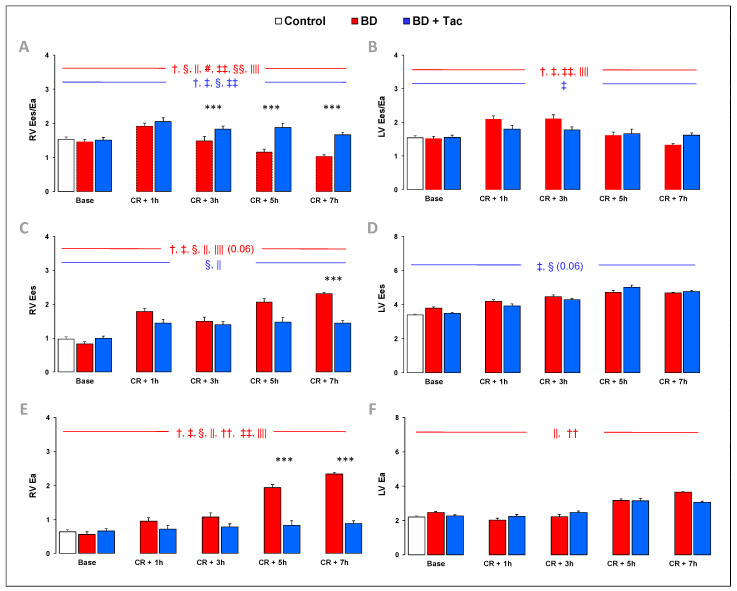
Right (**A**) and left (**B**) ventriculo-arterial coupling efficiency (Ees/Ea); right (**C**) and left (**D**) ventricular contractility (ventricular end-systolic elastance, Ees); and right (**E**) and left (**F**) ventricular afterload (effective pulmonary arterial elastance, Ea) were evaluated at baseline, 1 (CR + 1 h), 3 (CR + 3 h), 5 (CR + 5 h), and 7 h (CR + 7 h) after Cushing reflex (CR) in the control (*n* = 7; white bars), placebo (brain death, BD; *n* = 9; red bars), and tacrolimus pretreated brain death (BD + Tac; *n* = 8; blue bars) groups. Values are expressed as mean ± standard error of the mean. *** *p* < 0.05 BD vs. BD + Tac, † *p* < 0.05 Base vs. CR + 1 h, ‡ *p* < 0.05 Base vs. CR + 3 h, § *p* < 0.05 Base vs. CR + 5 h, || *p* < 0.05 Base vs. CR + 7 h, # *p* < 0.05 CR + 1 h vs. CR + 3 h, †† *p* < 0.05 CR + 1 h vs. CR + 5 h, ‡‡ *p* < 0.05 CR + 1 h vs. CR + 7 h, §§ *p* < 0.05 CR + 3 h vs. CR + 5 h, |||| *p* < 0.05 CR + 3 h vs. CR + 7 h. The comparisons between the different recording or sampling times are marked in red for the “BD” group and in blue for the “BD + Tac” group.

**Figure 3 ijms-24-10439-f003:**
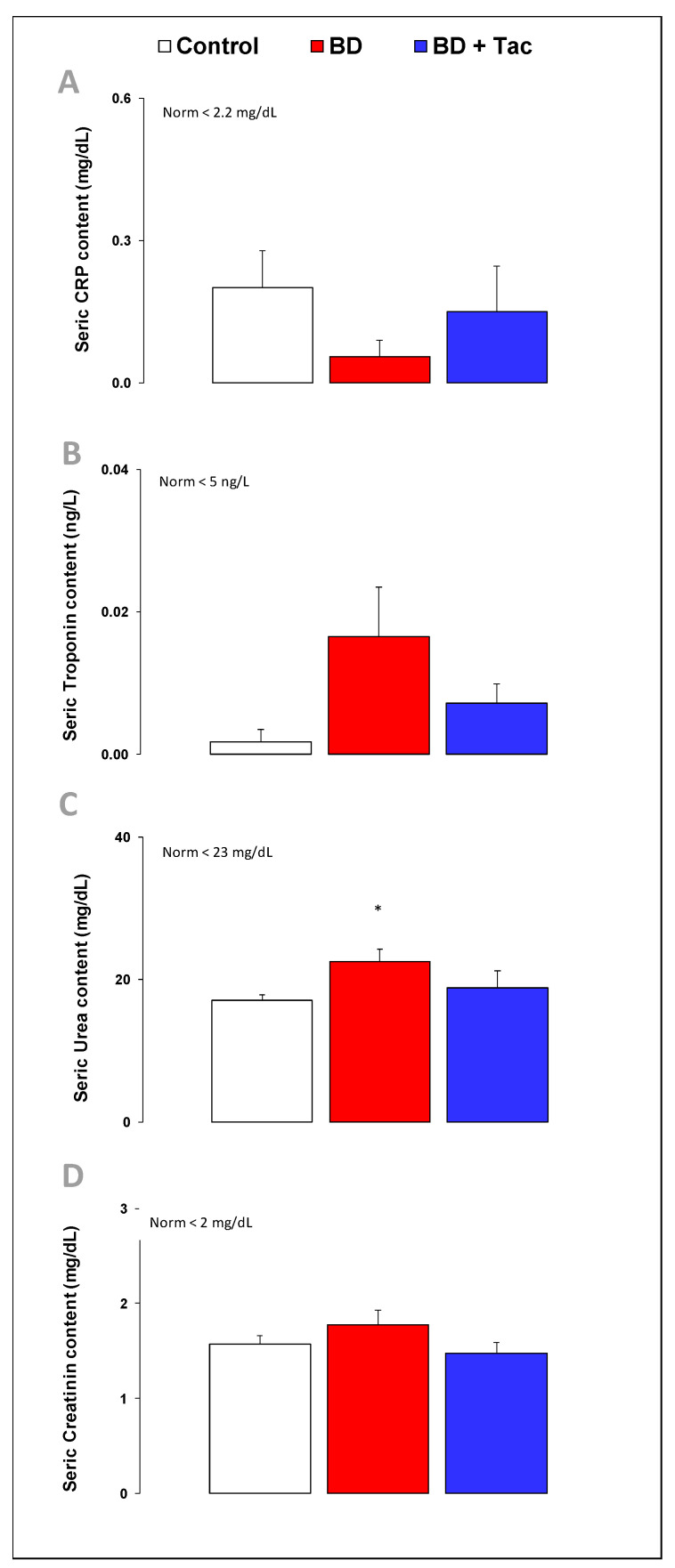
Serum C-reactive protein (**A**), troponin I (**B**), urea (**C**), and creatinine (**D**) concentrations in the control (*n* = 7; white bars), placebo (brain death, BD; *n* = 9; red bars), and tacrolimus pretreated brain death (BD + Tac; *n* = 8; blue bars) groups 7 h after the Cushing reflex (CR + 7 h). Values are expressed as mean ± standard error of the mean. * *p* < 0.05 control vs. BD.

**Figure 4 ijms-24-10439-f004:**
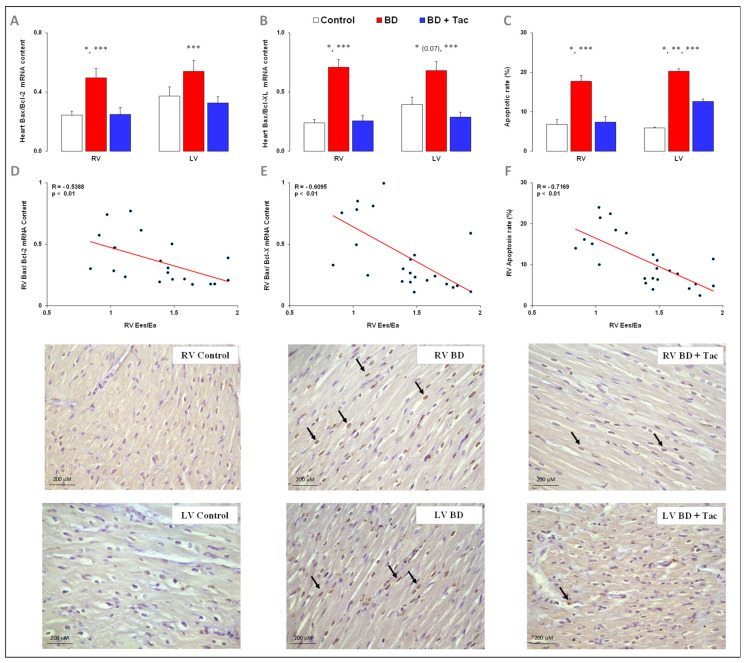
Right and left myocardial apoptosis in brain-death–induced right ventricular dysfunction with/without tacrolimus pretreatment. Pro-apoptotic Bax−to−Bcl−2 (**A**) and Bax−to−Bcl−XL (**B**) gene ratios in the right (RV) and left ventricles (LV) in the control (*n* = 7; white bars), placebo (brain death, BD; *n* = 9; red bars), and tacrolimus pretreated brain death (BD + Tac; *n* = 8; blue bars) groups. The cardiac apoptotic rate (**C**) (percentage) was evaluated as the ratio between the numbers of terminal deoxynucleotidyl transferase biotin-dUTP nick-end labeling (TUNEL)-positive cardiomyocytes (brown nuclei) and the total number of cardiomyocytes (brown + blue nuclei) in the RV and LV from the control (*n* = 7; white bars), placebo (brain death, BD; *n* = 9; red bars), and tacrolimus pretreated brain death (BD + Tac; *n* = 8; blue bars) groups 7 h after the Cushing reflex (CR + 7 h). Scale bars: 200 µm. Values are expressed as mean ± standard error of the mean. * *p* < 0.05 Control vs. BD, ** *p* < 0.05 Control vs. BD + Tac, *** *p* < 0.05 BD vs. BD + Tac. Correlations between the Ees/Ea ratio and the pro-apoptotic Bax−to−Bcl−2 (**D**) and Bax−to−Bcl−XL (**E**) ratios and apoptotic rate (**F**) in the RV. The arrows mark the terminal deoxynucleotidyl transferase biotin-dUTP nick-end labeling (TUNEL)-positive cardiomyocytes (brown nuclei).

**Figure 5 ijms-24-10439-f005:**
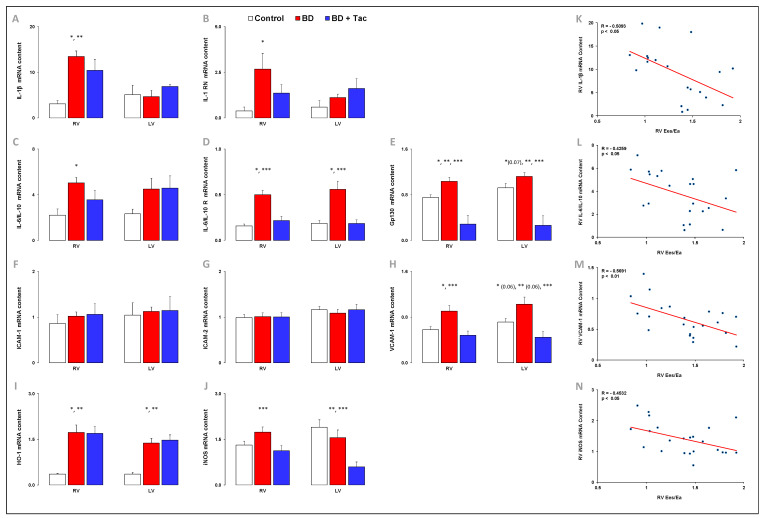
Myocardial expression of cytokines, adhesion molecules, heme oxygenase (HO) −1, and inducible nitric oxide synthase (iNOS) in brain-death–induced right ventricular dysfunction with/without tacrolimus pretreatment. The interleukin (IL) −1b (**A**), receptor for IL−1 (RN) (**B**), IL−6−to−IL−10 ratio (**C**), the ratio of the IL−6 receptor (R) to IL−10 R (**D**), receptor Gp130 for IL−6 (**E**), intercellular adhesion molecule (ICAM−1 (**F**), ICAM−2 (**G**)), vascular cell adhesion molecule (VCAM−1) (**H**), heme oxygenase (HO) −1 (**I**), and iNOS (**J**) relative mRNA content in the right (RV) and left ventricles (LV) from the control (*n* = 7; white bars), placebo (brain death, BD; *n* = 9; red bars), and tacrolimus pretreated brain death (BD + Tac; *n* = 8; blue bars) groups 7 h after the Cushing reflex (CR + 7 h). Values are expressed as mean ± standard error of the mean. * *p* < 0.05 Control vs. BD, ** *p* < 0.05 Control vs. BD + Tac, *** *p* < 0.05 BD vs. BD + Tac. Correlations between the Ees/Ea ratio and expression of IL−1β (**K**), IL−6/IL−10 (**L**), VCAM−1 (**M**), and iNOS (**N**) in the RV.

**Figure 6 ijms-24-10439-f006:**
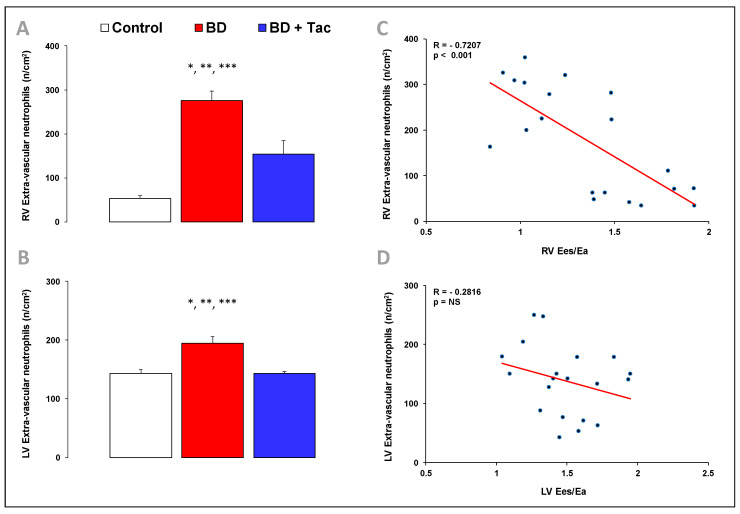
Characterization of inflammatory cells infiltrating the myocardium in brain-death-induced right ventricular dysfunction. The number of extravascular neutrophils (myeloperoxidase, MPO−positive cells) was determined in the right (RV) (**A**) and left ventricles (LV) (**B**) from the control (*n* = 7; white bars), placebo (brain death, BD; *n* = 9; red bars), and tacrolimus pretreated brain death (BD + Tac; *n* = 8; blue bars) groups 7 h after the Cushing reflex (CR + 7 h). Values are expressed as mean ± standard error of the mean. * *p* < 0.05 Control vs. BD, ** *p* < 0.05 Control vs. BD + Tac, *** *p* < 0.05 BD vs. BD + Tac. Correlations between the Ees/Ea ratio and the number of extravascular neutrophils in the right (**C**) and left (**D**) ventricles.

**Figure 7 ijms-24-10439-f007:**
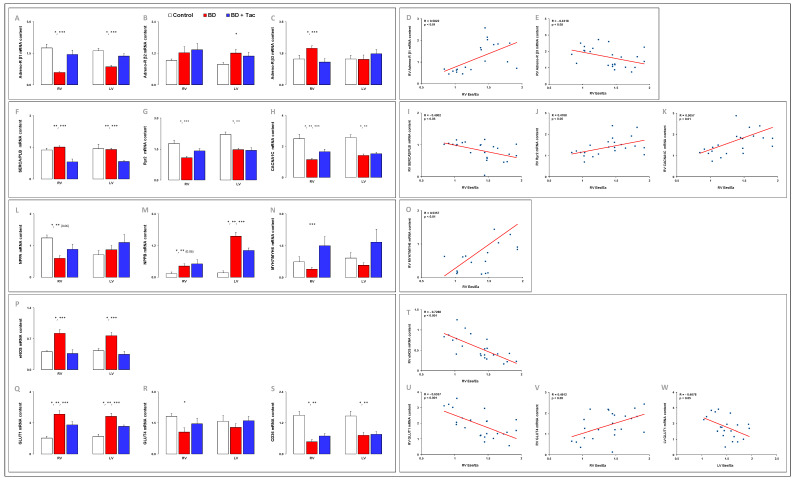
Myocardial expression of β-adrenergic receptors, regulators controlling calcium homeostasis, molecules regulating contractile function, natriuretic hormones, and signaling components involved in contraction-inducible substrate in brain-death–induced right ventricular dysfunction with/without tacrolimus pretreatment. b1 (**A**), 2 (**B**), and 3 (**C**) adrenergic receptors, sarcoplasmic/endoplasmic reticulum Ca^2+^ATPase 2a (SERCA2A)-to-phospholamban (PLB) ratio (**F**), ryanodine receptor type 2 (RyR2) (**G**), α1c subunit of the L-type calcium channel (CACNA1C) (**H**), natriuretic peptide A (NPPA) (**L**), brain natriuretic peptide (NPPB) (**M**), α-myosin heavy chain (MYH6) to β-myosin heavy chain (MYH7) ratio (**N**), endothelial (e)-NOS (**P**), glucose transporters (GLUT-) 1 (**Q**) and 4 (**R**), and thrombospondin receptor CD36 molecule (CD36) (**S**) relative mRNA content in right (RV) and left ventricles (LV) from the control (*n* = 7; white bars), placebo (brain death, BD; *n* = 9; red bars), and tacrolimus pretreated brain death (BD + Tac; *n* = 8; blue bars) groups 7 h after the Cushing reflex (CR + 7 h). Values are expressed as mean ± standard error of the mean. * *p* < 0.05 Control vs. BD, ** *p* < 0.05 Control vs. BD + Tac, *** *p* < 0.05 BD vs. BD + Tac. Correlations between the Ees/Ea ratio and gene expression for the β 1 (**D**) and β 3 (**E**) adrenergic receptors, SERCA2/PLB (**I**), Ryr2 (**J**), CACNA1C (**K**), MYH7/MYH6 (**O**), eNOS (**T**), GLUT1 (**U**), and GLUT4 (**V**) in the RV and GLUT1 in the LV (**W**).

**Figure 8 ijms-24-10439-f008:**
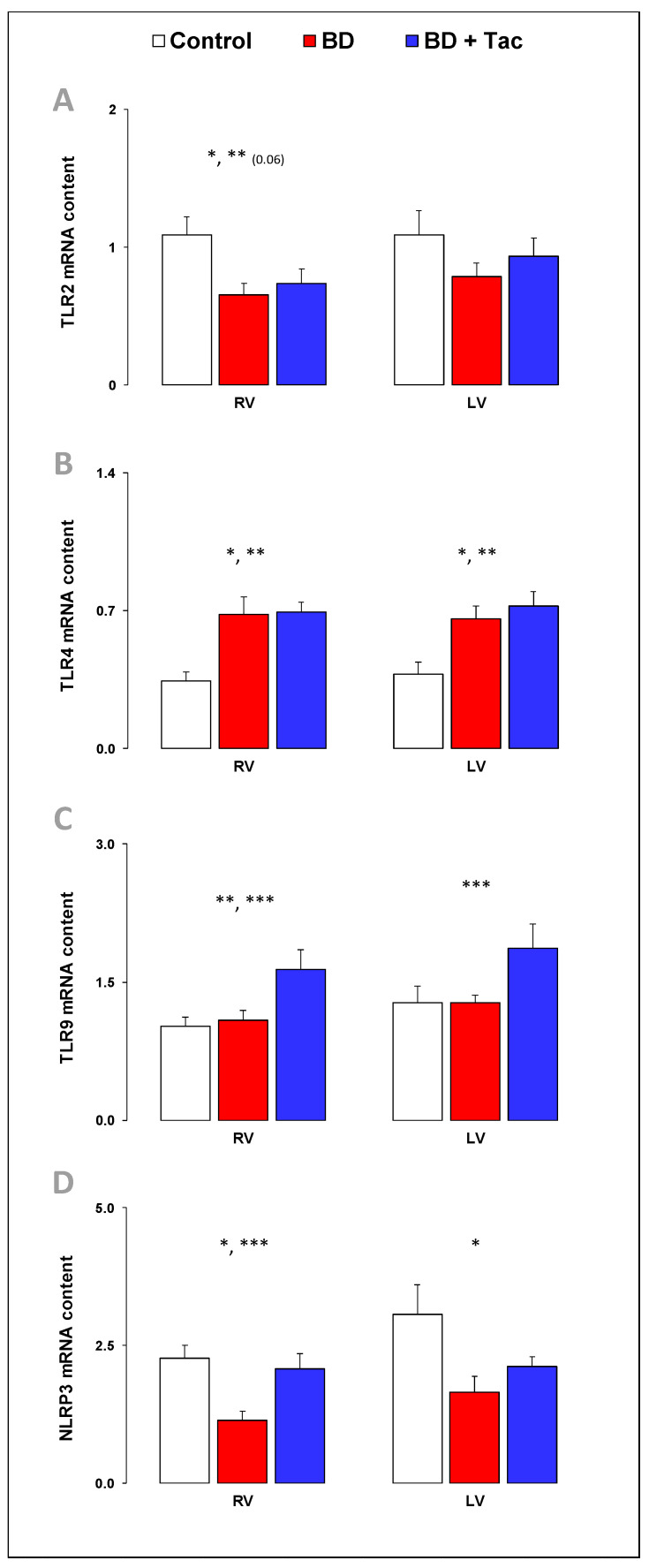
Myocardial expression of Toll-like receptor signaling pathways in brain-death–induced right ventricular dysfunction with/without tacrolimus pretreatment. Toll-like receptor (TLR-) 2 (**A**), 4 (**B**), and 9 (**C**), and NLR family pyrin domain containing 3 (NLRP3) (**D**) relative mRNA content in the right (RV) and left ventricles (LV) from the control (*n* = 7; white bars), placebo (brain death, BD; *n* = 9; red bars), and tacrolimus pretreated brain death (BD + Tac; *n* = 8; blue bars) groups 7 h after the Cushing reflex (CR + 7 h). Values are expressed as mean ± standard error of the mean. * *p* < 0.05 Control vs. BD, ** *p* < 0.05 Control vs. BD + Tac, *** *p* < 0.05 BD vs. BD + Tac.

**Table 1 ijms-24-10439-t001:** Primers used for real-time quantitative polymerase chain reaction (RTQ-PCR) in porcine myocardial tissue.

Gene	Primer Sequences
Bcl-2 associated X apoptosis regulator (Bax)	
*Sense*	5′-CGCATTGGAGATGAACTGG-3′
*Antisense*	5′-CGCCACTCGGAAAAAGACT-3′
B-CELL LYMPHOMA-2 (Bcl2)	
*Sense*	5′-GACTTTGCCGAGATGTCCAG-3′
*Antisense*	5′-ACAATCCTCCCCCAGTTCA-3′
Bcl-2-like 1 (BclXL)	
*Sense*	5′-TTGTGGCCTTTTTCTCCTTC-3′
*Antisense*	5′-CGATCCGACTCACCAATACC-3′
Interleukin-1beta (IL-1β)	
*Sense*	5′-CACCCAAAACCTGGACCTT-3′
*Antisense*	5′-TGCCTGATGCTCTTGTTC-3′
IL-1 receptor antagonist (IL1RN1)	
*Sense*	5′-GACTTTGCCGAGATGTCCAG-3′
*Antisense*	5′-ACAATCCTCCCCCAGTTCA-3′
Interleukin-6 (IL-6)	
*Sense*	5’-CCACCAGGAACGAAAGAGAG-3’
*Antisense*	5’-AGTAGCCATCACCAGAAGCAG-3’
IL-6 receptor (IL-6R)	
*Sense*	5′-CCGGAGGGAGACAACTCTTT-3′
*Antisense*	5′-GGCTGCAAGATTCCATAACC-3′
IL-6 signal transducer (Gp130)	
*Sense*	5’-ATGGCAGCGTACACAGATGA-3’
*Antisense*	5’-GCTAAACACACAGGCACGAC-3’
Interleukin-10 (*IL-10*)	
*Sense*	5’-TCATCAATTTCTGCCCTGTG-3’
*Antisense*	5’-TGTAGACACCCCTCTCTTGGA-3’
IL-10 receptor (IL-10R)	
*Sense*	5′-TTCAAGTCCGAGCGTTTCTT-3′
*Antisense*	5′-GGTTTCGTCATTGGTCGTCT-3′
Tumor necrosis factor-alpha (TNF-α)	
*Sense*	5′-TCTGGACTTTGCTGAATCTGG-3′
*Antisense*	5′-TGAGGGGGTCTGAAGGAGTAA-3′
Heme oxygenase-1 (HO-1)	
*Sense*	5’-CAGCATGCCCCAGGATTTGT-3
*Antisense*	5’-GACCTCGCCCTTCTGAAAGT-3’
NITRIC OXIDE-SYNTHASE 2 (NOS2 OR iNOS)	
*Sense*	5’-CTGCATGGATAAGTACAGGCTGACC-3
*Antisense*	5’-AGCTTCTGATCAATGTCATGAGCAA-3’
Intercellular adhesion molecule 1 (ICAM1)	
*Sense*	5′-ATTGTGAGGGGTGTCGAAGT-3′
*Antisense*	5′-TTCCCAGTTGTGTGTTTCCA-3′
Intercellular adhesion molecule 2 (ICAM2)	
Sense	5′-GGGCTCAGTGGAAGCTGTAT-3′
Antisense	5′-GGGAGAACACGCTGATGTTG-3′
Vascular cell adhesion molecule 1 (VCAM1)	
Sense	5′-GGAATTTACGTGTGCGAGGG-3′
*Antisense*	5′-TCCCTGGGAGCAACTTGAAC-3′
Adrenoceptor beta 1 (Adreno-R b1)	
*Sense*	5′-ACCCCAAGTGCTGCGATTT-3′
*Antisense*	5′-ATGCACAAGGGCACGTAGAA-3′
Adrenoceptor beta 2 (Adreno-R β2)	
*Sense*	5′-GATTCACAGGGGAGGAACTGTAG-3′
*Antisense*	5′-TTGTTTAGTGTTTGGCTGGGAG-3′
Adrenoceptor beta 3 (Adreno-R b3)	
*Sense*	5′-CAGAATGAGCCCTGTGGAGAT-3′
*Antisense*	5′-AGGTTGGTGAAAAGCCACTTG-3′
ATPase sarcoplasmic/endoplasmic reticulum Ca^2+^ transporting 2 (SERCA2A)	
*Sense*	5′-GGGAAAACCTTGCTGGAACT-3′
*Antisense*	5′-CTTCGCCTTCTTCAAACCAA-3′
Phospholamban (PLB)	
*Sense*	5′-AAACAGCCAAGGCTGCCTAAA-3′
*Antisense*	5′-GATACCAGGAAGGCAGGAAGC-3′
Ryanodine receptor 2 (Ryr2)	
Sense	5′-GTGAAGCAGCCCAAGGGTAT-3′
*Antisense*	5′-AAGGGACAGTGAGGCATTCG-3′
Ca^2+^ voltage-gated channel subunit alpha1 C (CACNA1C)	
*Sense*	5′-TTGACGCCTTGATTGTTGTG-3′
*Antisense*	5′-ATGGAGATGCGGGAGTTCT-3′
Sodium/calcium exchanger protein (NCX1)	
*Sense*	5′-ACTCTGGAATGCGGTTGGAG-3′
*Antisense*	5′-CCATGTAATGGAACATGTGTCTTCT-3′
Ca^2+^/calmodulin-dependent protein kinase II delta (CAMKIId)	
*Sense*	5′-GTGAAGCAGCCCAAGGGTAT-3′
*Antisense*	5′-AAGGGACAGTGAGGCATTCG-3′
Actin alpha 1 (ACTA1)	
*Sense*	5′-TCTATCGTCCACCGCAAAT-3′
*Antisense*	5′-CACTTGAGCAGATTCGTCGTC-3′
Myosin heavy chain 6 (MYH6)	
*Sense*	5′-CATCGGCGCCAAGCAAAAA-3′
*Antisense*	5′-AGAGTCTGGCGCTCATGTTT-3′
Myosin heavy chain 7 (MYH7)	
*Sense*	5′-CAAGGGCTTGAACGAGGAGTA-3′
*Antisense*	5′-TCCAGGACTGGGAGCTTTGT-3′
Natriuretic peptide A (NPPA)	
*Sense*	5′-TGTCCAATGCAGACCTGATG-3′
*Antisense*	5′-GGGGCATAGCCTCATCTTCT-3′
Natriuretic peptide B (NPPB)	
*Sense*	5′-AAGACGATGCGTGACTCTGG-3′
*Antisense*	5′-TACCTCCTGAGCACATTGCAG-3′
Nitric oxide synthase (NOS3 or eNOS)	
*Sense*	5’-CTTTCCTGTTGGCCTGACCA-3’
*Antisense*	5’-CCGGTTACTCAGACCCAAGG-3’
Solute carrier family 2 member 1 (SLC2A1 or GLUT1)	
*Sense*	5′-CCATTCAGACAAGCAACAGG-3′
*Antisense*	5′-TGGGATGTGGGTAAAGGAGA-3′
Solute carrier family 2 member 4 (SLC2A4 or GLUT4)	
*Sense*	5′-TTTCCAGTATGTTGCGGATG-3′
*Antisense*	5′-CGGGTTTCAGGCACTTTTAG-3′
CD36 molecule (CD36)	
*Sense*	5′-CAGCCATTTGTGGATACTTGG-3′
*Antisense*	5′-TGCTGGTTGGAATACAGTGG-3′
Toll-like receptor 2 (TLR2)	
*Sense*	5′-CTCTCGTTGCGGCTTCCAA-3′
*Antisense*	5′-TCCAGAGAGTTGACCTTGCAG-3′
Toll-like receptor 4 (TLR4)	
*Sense*	5′-CGTGCAGGTGGTTCCTAACAT-3′
*Antisense*	5′-TGACTGATGTGGGGATGTTGT-3′
NLR family pyrin domain containing 3 (NLRP3)	
*Sense*	5’-CAGGCTTCTGGGACACCTTT-3’
*Antisense*	5’-CAGGCTTCTGGGACACCTTT-3’
RIBOSOMAL PROTEIN-L4 (RPL4)	
*Sense*	5′-AAACCAAGGAGGCTGTTCTG-3′
*Antisense*	5′-CATTCGCTGAGAGGCATAAA-3′

## Data Availability

The data sets and materials generated and/or analyzed during the current study are available from the corresponding author (asmae.belhaj@chuuclnamur.uclouvain.be) on reasonable request.

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
