# Peer review of "Beneficial Effects of Tacrolimus on Brain-Death-Associated Right Ventricular Dysfunction in Pigs"

_ijms, 2023, doi:10.3390/ijms241310439_

Round 1
Reviewer 1 Report
The manuscript describes effect of tacrolimus administration on brain death associated right ventricular dysfunction in a porcine model
Figures in landscape - suggest that in formatting for final paper if these could be in portrait rather than landscape it would be much easier to follow the different graphs that make up the figures
Figure 3 onwards - legends do not appear to follow the figure -the remaining figures and legends also appear to be out of sync.
A thorough analysis at mRNA level for several genes involved in inflammatory responses - it would now be useful to follow up with protein expression for those markers that have shown significant differences in expression
some typos and formatting to be resolved
Author Response
Dear Editor,
Dear Reviewers,
We revised our manuscript entitled “BENEFICIAL EFFECTS OF CALCINEURIN INHIBITION ON BRAIN DEATH-ASSOCIATED RIGHT VENTRICULAR DYSFUNCTION IN PIGS” following the comments of the reviewers.
We thank the editor and the reviewers for their suggestions and comments, which have improved the presentation and clarity of the manuscript. The figures have been reworked and the title of the manuscript changed in line with the comments of the reviewers. More information about the validity and the pertinence of the present experimental model of brain death in pigs has been added. The manuscript has been thoroughly proofread for typos and formatting problems.
We offer the following point-by-point reply to reviewers’comments.
Reviewer#1:
- Figures in landscape - suggest that in formatting for final paper if these could be in portrait rather than landscape it would be much easier to follow the different graphs that make up the figures
R1: We thank the reviewer for this suggestion. Figures 1, 2, 3, 4, 6 and 8 have been adapted to portrait format accordingly. However, to ensure optimum legibility of Figures 5 and 7, we suggest keeping the latter two in landscape format. Figures 5 and 7 are more complex to rearrange.
- Figure 3 onwards - legends do not appear to follow the figure -the remaining figures and legends also appear to be out of sync.
R2: We thank the reviewer for pointing this out and apologize for the inconvenience. It has been modified accordingly.
- A thorough analysis at mRNA level for several genes involved in inflammatory responses - it would now be useful to follow up with protein expression for those markers that have shown significant differences in expression
R3: Agreed. We have previously reported, in a similar experimental model of brain death in pigs, that the expression of inflammatory mediators at gene and protein levels evolved in a similar way in cardiac and pulmonary tissues (Belhaj, Dewachter et al. J Heart Lung Transplant. 2016; 35: 1505-1518; Belhaj, Dewachter et al. PLoS One. 2017 ; 12: e0181899). We also recently showed in this experimental model that circulating levels of proinflammatory cytokines, including the IL-6-to-IL-10 ratio, IL-1a and IL-1b were increased 7 hours after brain death (Belhaj, Dewachter L et al. Am J Respir Crit Care Med. 2022; 206: 584-595). Finally, it remains difficult to find reliable, good-quality antibodies that work in pig samples for Western Blotting experiments.
Therefore, and because the time available to review the present manuscript is short, we hypothesized that the mRNA expression profile could be used to reflect adequately the pathobiological profile within a limited follow-up time (5 to 7 hours) after Cushing's reflex.
Comments on the Quality of English Language - some typos and formatting to be resolved
R: The manuscript has been thoroughly proofread. We hope that the typos and formatting problems have now been solved.
Our revised manuscript includes:
- Point-by-point responses to reviewer comments and to any issues raised in this letter.
- A "Marked up" version of your previous manuscript text (Use red font to indicate the revised portions of the manuscript) followed by the figures for the marked-up version.
- A "Clean" version of your revised manuscript text (use black fonts, no red marked-up fonts) followed by the complete set of figures for the clean version.
We respectfully hope that this revised version of our manuscript might be found acceptable.
We thank the reviewers for their valid comments and criticisms which were very helpful to improve our manuscript.
Yours sincerely,
Asmae Belhaj, MD, PhD
Laurence Dewachter, PharmD, PhD
Benoit Rondelet, MD, PhD
Reviewer 2 Report
Please provide more background about the clinical significance of this BD model as well as the BD-induced RV dysfunction.
Please provide more comprehensive background concerning the known roles of calcineurin in PH and RV dysfunction.
To strengthen the notion that "calcineurin inhibition" prevents RV dysfunction induced by BD, please test other calcineurin inhibitors.
Author Response
Dear Reviewer,
We revised our manuscript entitled “BENEFICIAL EFFECTS OF CALCINEURIN INHIBITION ON BRAIN DEATH-ASSOCIATED RIGHT VENTRICULAR DYSFUNCTION IN PIGS” following the comments of the reviewers.
We thank thethe reviewer for his suggestions and comments, which have improved the presentation and clarity of the manuscript. The figures have been reworked and the title of the manuscript changed in line with the comments of the reviewers. More information about the validity and the pertinence of the present experimental model of brain death in pigs has been added. The manuscript has been thoroughly proofread for typos and formatting problems.
We offer the following point-by-point reply to reviewers’comments.
Reviewer#2:
- Please provide more background about the clinical significance of this BD model as well as the BD-induced RV dysfunction.
R1: We thank the reviewer for this comment. Transplantation is the gold standard treatment for patients with end-stage heart failure but ever-increasing disparity between available organs and potential recipients is a cause of avoidable morbidity and mortality (1). Ongoing efforts are being made to increase the quantity and quality of organs available for transplant. Significant brain injury of any aetiology will cause a systemic response (2), creating a proinflammatory environment prior to the occurrence of brain death. Brain death itself is responsible for a variety of upregulation of in situ inflammatory processes and of cell damage-associated signaling pathways (3-4), neuro-endocrine modulations (5), haemodynamic changes (6), apoptosis induction (7), and increased release of inflammatory cytokines and stress hormones (8), which can induce direct and indirect adverse functional sequelae in right ventricle.
Brain death alters the physiologic dialogue between the heart and lungs. So, in the neuro-cardiac theory, the catecholamine storm accompanying the Cushing’s reflex leads to a direct toxic myocardial injury leading to right ventricular dysfunction (9). While in the neuro-hemodynamic or the blast proposition, the rapid severe increase in hydrostatic pressures following the catecholamine surge results in a net shift of blood volume from the systemic to the pulmonary circulation leading to the development of transudative pulmonary edema and pulmonary hypertension leading to right heart dysfunction (10).
There is a consensus that a better characterization and quantification of a single “primum movens” of brain death induced right heart injury would be of therapeutic relevance with the goal of improving early and long-term graft function after transplantation. In that way, animal models can fill gaps in knowledge that are unattainable in clinical settings (11). Our model mimicking cerebral hemorrhage is associated with right ventricular-arterial uncoupling related to the Cushing reflex-associated catecholamine myocarditis and inflammatory and pro-apoptotic changes (12). Neuro-humoral activation, apoptosis as well as up-regulation of pro-inflammatory cytokines are accompagnied with increased myocardial expression of endothelial adhesion molecules, and myocardial infiltration by neutrophils, which directly influence the heart failure process. All these factors contributing to poor transplant outcomes, but the specific role played by the immunity in the early brain death has received less attention so far (13) justifying the aim of the present study.
This is now better explained and specified in the Introduction section (see pages 1-2, lines 7-41).
- Banner NR, Rogers CA, Bonser RS, et al. Effect of heart transplantation on survival in ambulatory and decompensated heart failure. Transplantation. 2008;96(11, article 8).
- Lee S-T, Chu K, Jung KH, et al. Cholinergic anti-inflammatory pathway in intracerebral hemorrhage. Brain Research. 2010;1309:164–171.
- Anyanwu AC, Banner NR, Radley-Smith R, Khaghani A, Yacoub MH. Long-term results of cardiac transplantation from live donors: the domino heart transplant. Journal of Heart and Lung Transplantation. 2002;21(9):971–975.
- Khaghani A, Birks EJ, Anyanwu AC, Banner NR. Heart transplantation from live donors: ‘Domino procedure’ Journal of Heart and Lung Transplantation. 2004;23(9, supplement 1):S257–S259.
- Sousa KS, Quiles CL, Muxel SM, Trevisan IL, Ferreira ZS, Markus RP. Brain Damage-linked ATP Promotes P2X7 Receptors Mediated Pineal N-acetylserotonin Release. Neuroscience. 2022 Sep 1;499:12-22.
- Wells MA, See Hoe LE, Molenaar P, Pedersen S, Obonyo NG, McDonald CI, Mo W, Bouquet M, Hyslop K, Passmore MR, Bartnikowski N, Suen JY, Peart JN, McGiffin DC, Fraser JF; Dead Heart Project. Compromised right ventricular contractility in an ovine model of heart transplantation following 24 h donor brain stem death. Pharmacol Res. 2021 Jul;169:105631.
- Marasco SF, Arthur JF, Ou R, Bailey M, Rosenfeldt F. Apoptotic Markers in Donor Hearts After Brain Death vs Circulatory Death. Transplant Proc. 2021 Mar;53(2):612-619.
- Zhang YH, Lin JX, Vilcek J. Interleukin-6 induction by tumor necrosis factor and interleukin-1 in human fibroblasts involves activation of a nuclear factor binding to a kappa B-like sequence. Mol Cell Biol. 1990;10:3818–23.
- Connor RC. Myocardial damage secondary to brain lesions. Am Heart J 1969;78:145–148.
- Sarnoff SJ, Sarnoff LC. Neurohemodynamics of pulmonary edema: II. The role of sympathetic pathways in the elevation of pulmonary and stemic vascular pressures following the intracisternal injection of fibrin. Circulation 1952;6:51–62.
- See Hoe LE, Wells MA, Bartnikowski N, Obonyo NG, Millar JE, Khoo A, Ki KK, Shuker T, Ferraioli A, Colombo SM, Chan W, McGiffin DC, Suen JY, Fraser JF. Heart Transplantation From Brain Dead Donors: A Systematic Review of Animal Models. Transplantation. 2020 Nov;104(11):2272-2289.
- Belhaj A, Dewachter L, Rorive S, Remmelink M, Weynand B, Melot C, Galanti L, Hupkens E, Sprockeels T, Dewachter C, Creteur J, McEntee K, Naeije R, Rondelet B. Roles of inflammation and apoptosis in experimental brain death-induced right ventricular failure. J Heart Lung Transplant. 2016 Dec;35(12):1505-1518.
- Adrie C, Monchi M, Fulgencio JP, Cottias P, Haouache H, Alvarez-Gonzalvez A, et al. Immune status and apoptosis activation during brain death. Shock 2010;33:353–362.
- Please provide more comprehensive background concerning the known roles of calcineurin in PH and RV dysfunction.
R2: Pulmonary hypertension and right heart dysfunction are characteristic elements of hemodynamic disturbances associated with brain death (1).
The calcineurin signaling is activated and involved in the physiopathology of pulmonay hypertension (2-3) and the associated right ventricular dysfunction (4). This is why calcineurin inhibitors have been studied as a potential treatment for pulmonary arterial hypertension (PAH) in experimental models (5-6). Low-dose tacrolimus reversed pulmonary vascular remodeling and improved right ventricular (RV) function by restoring BMP signaling in two different experimental models of pulmonary hypertension (7), stabilized three patients with end-stage PAH (8), and was well-tolerated and safe when tested in a larger cohort of patients with stable PAH (9). Recently, tacrolimus has been shown to reduce RV fibrosis in a BMPs signaling pathway-dependent manner, preserve RV vasculature, and improve RV function over time (10).
Mechanistically, tacrolimus binds the 12-kDa FK506-binding protein (FKBP12) and thereby it has been shown to activate downstream BMP signaling pathway in pulmonary artery endothelial cells (7). Tacrolimus requires the activin receptor-like kinase 1 (ALK1) as a co-receptor to reduce collagen production in cardiac fibroblasts. Interestingly, FKBP12 itself is linked to the calcium release channel of cardiac muscle, and pharmacological dissociation of this complex alters its gating characteristics (11-12), leading to increased calcium accumulation in cardiomyocytes and a potential additional positive inotropic effect associated with tacrolimus treatment (13-14).
This is now better explained in the Discussion section (see page 17-18, lines 395-401).
- Belhaj A, Dewachter L, Rorive S, Remmelink M, Weynand B, Melot C, Galanti L, Hupkens E, Sprockeels T, Dewachter C, Creteur J, McEntee K, Naeije R, Rondelet B. Roles of inflammation and apoptosis in experimental brain death-induced right ventricular failure. J Heart Lung Transplant. 2016 Dec;35(12):1505-1518.
- He RL, Wu ZJ, Liu XR, Gui LX, Wang RX, Lin MJ. Calcineurin/NFAT Signaling Modulates Pulmonary Artery Smooth Muscle Cell Proliferation, Migration and Apoptosis in Monocrotaline-Induced Pulmonary Arterial Hypertension Rats. Cell Physiol Biochem. 2018;49(1):172-189.
- de Frutos S, Nitta CH, Caldwell E, Friedman J, González Bosc LV. Regulation of soluble guanylyl cyclase-alpha1 expression in chronic hypoxia-induced pulmonary hypertension: role of NFATc3 and HuR. Am J Physiol Lung Cell Mol Physiol. 2009 Sep;297(3):L475-86.
- Qin N, Gong QH, Wei LW, Wu Q, Huang XN. Total ginsenosides inhibit the right ventricular hypertrophy induced by monocrotaline in rats. Biol Pharm Bull. 2008 Aug;31(8):1530-5.
- Koulmann N, Novel-Chaté V, Peinnequin A, Chapot R, Serrurier B, Simler N, Richard H, Ventura-Clapier R, Bigard X. Cyclosporin A inhibits hypoxia-induced pulmonary hypertension and right ventricle hypertrophy. Am J Respir Crit Care Med. 2006 Sep 15;174(6):699-705.
- Wang YX, Reyes-García J, Di Mise A, Zheng YM. Role of ryanodine receptor 2 and FK506-binding protein 12.6 dissociation in pulmonary hypertension. J Gen Physiol. 2023 Mar 6;155(3):e202213100. doi: 10.1085/jgp.202213100.
- Spiekerkoetter E, Tian X, Cai J, Hopper RK, Sudheendra D, Li CG, et al. FK506 activates BMPR2, rescues endothelial dysfunction, and reverses pulmonary hypertension. J Clin Invest. 2013;123:3600–3613.
- Spiekerkoetter E, Sung YK, Sudheendra D, Bill M, Aldred MA, van de Veerdonk MC, et al. Low-dose FK506 (tacrolimus) in end-stage pulmonary arterial hypertension. Am J Respir Crit Care Med. 2015;192:254–257.
- Spiekerkoetter E, Sung YK, Sudheendra D, Scott V, Del Rosario P, Bill M, et al. Randomised placebo-controlled safety and tolerability trial of FK506 (tacrolimus) for pulmonary arterial hypertension. Eur Respir J. 2017;50:1602449.
- Boehm M, Tian X, Ali MK, Mao Y, Ichimura K, Zhao M, Kuramoto K, Dannewitz Prosseda S, Fajardo G, Dufva MJ, Qin X, Kheyfets VO, Bernstein D, Reddy S, Metzger RJ, Zamanian RT, Haddad F, Spiekerkoetter E. Improving Right Ventricular Function by Increasing BMP Signaling with FK506. Am J Respir Cell Mol Biol. 2021 Sep;65(3):272-287.
- Mayrleitner M, Timerman AP, Wiederrecht G, Fleischer S. The calcium release channel of sarcoplasmic reticulum is modulated by FK-506 binding protein: effect of FKBP-12 on single channel activity of the skeletal muscle ryanodine receptor. Cell Calcium. 1994;15:99–108.
- Ahern GP, Junankar PR, Dulhunty AF. Single channel activity of the ryanodine receptor calcium release channel is modulated by FK-506. FEBS Lett. 1994;352:369–374.
- Hemnes AR, Brittain EL, Trammell AW, Fessel JP, Austin ED, Penner N, et al. Evidence for right ventricular lipotoxicity in heritable pulmonary arterial hypertension. Am J Respir Crit Care Med. 2014;189:325–334.
- Yano M, Kobayashi S, Kohno M, Doi M, Tokuhisa T, Okuda S, Suetsugu M, Hisaoka T, Obayashi M, Ohkusa T, Kohno M, Matsuzaki M. FKBP12.6-mediated stabilization of calcium-release channel (ryanodine receptor) as a novel therapeutic strategy against heart failure. Circulation. 2003 Jan 28;107(3):477-84.
- To strengthen the notion that "calcineurin inhibition" prevents RV dysfunction induced by BD, please test other calcineurin inhibitors.
R3: We thank the reviewer for this comment.
When we began the present research project, we planned to test different calcineurin inhibitors to compare their effects and submitted the project at the local ethics committee of the Université Libre de Bruxelles. This project was rejected as initially proposed, as it required the inclusion of a large number of animals to obtain reliable results with potentially minor differences between the different tested calcineurin-inhibiting drugs. Indeed, tacrolimus and cyclosporine differ in their chemical structure: cyclosporine is a cyclic endecapeptide (1), whereas tacrolimus is a macrocyclic lactone (2). However, they act in a similar manner, both are important calcineurin phosphatase inhibitors (3). The pharmacogenetics of tacrolimus and cyclosporine is complex, and a great number of factors likely contribute to its variability (4).
Moreover, adding now an extra arm to the current study design would considerabily alter the power of the statistical study and make randomization inadequate.
Thus, in accordance with your comment and to fit better with the results of the present study, we propose to present this work as testing the specific effects of “tacrolimus” rather than of “calcineurin inhibitors”. This has been accordingly modified, including in the Title.
- Fahr A. Cyclosporin clinical pharmacokinetics. Clin Pharmacokinet. 1993;24:472–495.
- Organ Procurement and Transplantation Network (OPTN) and Scientific Registry of Transplant Recipients (SRTR) OPTN/SRTR 2010 Annual Data Report. Vol. 2012 Department of Health and Human Services, Health Resources and Services Administration, Healthcare Systems Bureau, Division of Transplantation; 2011.
- Kapturczak MH, Meier-Kriesche HU, Kaplan B. Pharmacology of calcineurin antagonists. Transplant Proc. 2004;36:25S–32S.
- Barbarino JM, Staatz CE, Venkataramanan R, Klein TE, Altman RB. PharmGKB summary: cyclosporine and tacrolimus pathways. Pharmacogenet Genomics. 2013 Oct;23(10):563-85.
Our revised manuscript includes:
- Point-by-point responses to reviewer comments and to any issues raised in this letter.
- A "Marked up" version of your previous manuscript text (Use red font to indicate the revised portions of the manuscript) followed by the figures for the marked-up version.
- A "Clean" version of your revised manuscript text (use black fonts, no red marked-up fonts) followed by the complete set of figures for the clean version.
We respectfully hope that this revised version of our manuscript might be found acceptable.
We thank the reviewer for his valid comments and criticisms which were very helpful to improve our manuscript.
Yours sincerely,
Asmae Belhaj, MD, PhD
Laurence Dewachter, PharmD, PhD
Benoit Rondelet, MD, PhD
Round 2
Reviewer 2 Report
.